# Seroprevalence and preventive practices of dengue and chikungunya among school children in Bangkok: Gaps in prevention and vaccination strategies

Thitiya Yakasaem[1], Thidarat Jupimai[2], Nattapong Jitrungruengnij[2,3], Napaporn Chantasrisawad[1,2,4], Ekasit Kowitdamrong[5], Padet Siriyasatien[6], Sunthorn Sunthornchart[7], Nattinee Isarankura Na Ayudaya[8], Paveena Angkhananukit[9], Pitsamai Ruansil[10], Kanchana Nakhapakorn[11], Eric Daudé[12], Alexandre Cebeillac[12], Richard Paul[13], Thanyawee Puthanakit[1,2], Watsamon Jantarabenjakul[1]*

1 Division of Pediatric Infectious Disease, Department of Pediatrics, Faculty of Medicine, Chulalongkorn University, Bangkok, Thailand, 2 Center of Excellence for Pediatric Infectious Diseases and Vaccines Faculty of Medicine, Chulalongkorn University, Bangkok, Thailand, 3 Department of Pediatrics, Charoenkrung Pracharak Hospital, Medical Service Department, Bangkok Metropolitan Administration, Bangkok, Thailand, 4 Thai Red Cross Emerging Infectious Disease Clinical Center, King Chulalongkorn Memorial Hospital, Bangkok, Thailand, 5 Department of Microbiology, Faculty of Medicine, Chulalongkorn University, Bangkok, Thailand, 6 Center of Excellence in Vector Biology and Vector Borne Disease, Department of Parasitology, Faculty of Medicine, Chulalongkorn University, Bangkok, Thailand, 7 Bangkok Metropolitan Administration, Bangkok, Thailand, 8 Division of Communicable Disease Control, Health Department, The Bangkok Metropolitan Administration (BMA), Bangkok, Thailand, 9 Public Health Service Center 24, Bangkok, Thailand, 10 Department of Education, Bangkok, Thailand, 11 Faculty of Environmental and Resource Studies, Mahidol, Bangkok, Thailand, 12 Institut de recherche sur l'Asie du Sud-Est contemporaine IRASEC, CNRS, Bangkok, Thailand, 13 Institut Pasteur, Université Paris Cité, CNRS UMR 2000, INRAE USC 1510, Ecology and Emergence of Arthropod-borne Pathogens Unit, Paris, France

* Watsamon.J@chula.ac.th

## Abstract

### Background

Dengue and chikungunya, both transmitted by *Aedes* mosquitoes, continue to pose significant public health concerns in Thailand, particularly during the rainy season. Despite ongoing vector control efforts, the incidence of infection remains high, with an increasing trend observed in chikungunya. This underscores the need for additional control measures, including vaccination, to reduce disease burden and morbidity. This study aims to assess the seroprevalence of dengue and chikungunya infections among children aged 10–15 years in Bangkok and to evaluate the knowledge, attitudes, and practices (KAP) related to mosquito-borne disease prevention.

### Methodology

A cross-sectional descriptive study was conducted across 12 schools in Bangkok. Children aged 10–15 years were included. Seroprevalence was determined using rapid diagnostic tests (Abbott DENGUE IgG/IgM and Citest Chikungunya IgG/IgM)

**Data availability statement:** All relevant data are within the manuscript and its Supporting information files.

**Funding:** This research was funded by the Ratchadapiseksomphot Fund and the Ratchadapiseksomphot Matching Fund, Faculty of Medicine, Chulalongkorn University (grant numbers RA67/027 and RA-MF20/67 to WJ) (https://grad.md.chula.ac.th/scholarship_department_detail.php?s=22). The study was also supported by Institut Pasteur, France through a grant from the Agence Nationale de la Recherche (ANR-19-CE03-0004-02 to ED and RP). The funders had no role in study design, data collection and analysis, decision to publish, or preparation of the manuscript. No authors received salary support from this work.

**Competing interests:** The authors have declared that no competing interests exist.

based on the immunochromatography technique, using fingertip blood samples. Parents completed KAP questionnaires, including factors influencing vaccination decisions.

### Principal findings

From June to August 2024, 937 participants were enrolled, with a mean (SD) age of 11 (1.6) years; 67% were aged 10–12 years, and 33% were aged 13–15 years. The seroprevalence of dengue was 28.1% (95% CI 25.2-31.0), while that of chikungunya was 6.3% (95% CI 4.7-7.9). KAP assessments revealed a high level of awareness regarding symptoms and transmission; however, notable deficiencies in preventive behaviors were identified. Only 14.8% of respondents reported consistent use of mosquito repellent, and 17.5% routinely inspected and removed mosquito larvae from their homes.

### Conclusion

The substantial seroprevalence of dengue and the emerging trend of chikungunya among children in Bangkok highlights the urgent need to enhance community education and strengthen vector control interventions. Expanding dengue vaccination coverage and raising awareness about chikungunya prevention, including consideration for future vaccine implementation, are essential to mitigating future outbreaks and reducing the disease burden.

### Trial registration

Thai Clinical Trials under the number TCTR20240404002 (https://www.thaiclinicaltrials.org/show/TCTR20240404002).

### Author summary

Mosquito-borne diseases, including dengue and chikungunya, are significant health problems in Southeast Asia. In our study, we evaluate the seroprevalence of these diseases using rapid blood tests in school-based settings among children in Bangkok. The seroprevalence data highlight the risk of exposure, particularly among children, and can guide preventive practices, including vaccination. Despite the widespread recognition of these diseases, preventive practices remain limited. Therefore, implementing effective preventive measures and vaccination strategies could significantly reduce the severity of infections and improve public health outcomes.

### Introduction

Mosquito-borne diseases, particularly dengue and chikungunya, remain significant public health concerns in Southeast Asia. As of October 2024, the reported incidence

of dengue was 156 cases per 100,000 population, with Bangkok recording a morbidity rate of 115 per 100,000 [1]. In 2023, the highest infection rates were observed in children and adolescents [2]. A systematic review reported seroprevalence among Thai children aged 2–17 years, assessed by the plaque reduction neutralization test (PRNT), reaching up to 80% in certain regions [3]. The seroprevalence increases with age; among 2–4-year-olds, reports indicate 48%, 5–8 years report 61.9%, 9–12 years report 79.5% and 13–16 year report 84.2% [4]. Chikungunya, while less prevalent than dengue, is more commonly reported in adolescents and adults, with symptoms including rash, severe joint pain, and potential neurological complications in children [5]. A systematic review study covering the African, American, and Southeast Asian regions found a seroprevalence of 7% in children under five years old, consistent with findings from a 2014–2015 study [6,7]. Chikungunya outbreaks have been more frequent in Thailand in southern regions. A study in 2017 reported a seropositivity rate of Chikungunya IgG antibodies by commercial enzyme-linked immunosorbent assays (ELISA) of 3% among individuals aged 10–19 in central Thailand, compared to 20% in the South [8]. However, data on Chikungunya seroprevalence in Thai children remains limited.

Given that *Aedes* mosquitoes transmit both diseases, their spread is influenced by social and environmental factors. While prevention measures such as vector control and personal protection are widely promoted, adherence to these practices varies. Effective disease prevention relies on public knowledge, attitudes, and practices (KAP). A cross-sectional study in the South of Thailand found that children previously infected with dengue exhibited better preventive practices, and over 70% of children cited teachers as their primary source of dengue information [9]. Parental engagement in vector control efforts may further strengthen prevention efforts, and access to vaccination and other interventions also impacts disease control efforts. Despite ongoing vector control measures and the introduction of two dengue vaccines, Dengvaxia in 2017 and Qdenga in 2023, dengue incidence continues to rise, particularly after the COVID-19 pandemic. The WHO recommends dengue vaccination in high-prevalence areas [10]. This is especially so for Dengvaxia, implementation of which is advised only for individuals having had a natural first time exposure to the virus [11]. Currently, the Chikungunya vaccine is available when there is an outbreak or when there is evidence of viral transmission within the past five years, especially in older adults with co-morbidities [12]. This study aims to determine the seroprevalence of dengue and chikungunya virus infections among children aged 10–15 years in Bangkok and to assess their parents' KAP related to mosquito-borne disease prevention. Understanding both the immunological exposure and behavioral factors in this age group is critical for informing targeted public health strategies. The study is based on the following hypotheses: (1) a substantial proportion of children have serological evidence of prior dengue and/or chikungunya virus infection, and (2) higher levels of knowledge, positive attitudes, and appropriate preventive practices are associated with a lower likelihood of seropositivity. These objectives and hypotheses are intended to guide both data interpretation and the development of future school- and community-based interventions.

## Materials and methods

### Ethics statement

Written assent was obtained from participants whose parents provided informed consent for blood sampling. Parents who consented to participate were included in the questionnaire-based survey. This study was part of the Geomosquito study and was approved by the Institut Pasteur Institutional Review Board (N°2023–136), the Chulalongkorn Institutional Review Board (IRB No. 0648/66), and the Bangkok Metropolitan Administration (BMA) Human Research Ethics Committee (BMAHREC No. E005hc/67). It was registered with the Thai Clinical Trials under the number TCTR20240404002 (https://www.thaiclinicaltrials.org/show/TCTR20240404002).

### Study design and inclusion/exclusion criteria

A cross-sectional observational study was conducted in 12 schools in Bangkok. The study included children aged 10–15. Children with medical conditions that could interfere with blood sampling on the study day were excluded.

## Study procedures

The Bangkok metropolitan area was divided into five regions: Northern, Southern, Eastern, Western, and Central. Schools were selected from each region after initial contact and agreement to participate. The study was conducted on school premises, where parents provided consent for their children's participation and completed questionnaires to screen history and assess their knowledge, attitudes, practices, and factors related to dengue and chikungunya vaccination. Students received study details, assent, and parent consent documents, and they participated in an educational session about dengue diseases and prevention (Fig 1). Blood samples were collected via fingertip capillary blood sampling for dengue and chikungunya antibody testing.

## Dengue and chikungunya screening test

Rapid diagnostic tests were used to detect antibodies. The Abbott DENGUE IgG/IgM (Abbott Laboratories, USA) was used for dengue, while the Citest IgG/IgM (CITEST diagnostic inc., Canada) was used for chikungunya. Both tests utilize immunochromatography, requiring only 10 microlites for IgG and IgM detection. The reported sensitivity and specificity for the dengue test are 94.2% and 96.4%, while for the chikungunya test, they are 90.3% and 99.9% for diagnosing acute illness, respectively [13,14]. Blood samples were collected using a capillary tube and transferred to the test kit with buffer, and results were available in 15–20 minutes. Two trained technicians independently interpreted the results, with a third confirmation if the results were discordant.

## Questionnaire development

A structured questionnaires was used to collected demographic data and assess parents' knowledge, attitudes, practices, and factors influencing vaccination uptake for dengue and chikungunya. The questionnaire consisted of 28 questions and was designed to take 3–5 minutes to complete. The questionnaire was reviewed by physicians to ensure content validity, and its reliability was assessed using Cronbach's alpha.

## Sample size calculation

The sample size was determined based on expected seroprevalence rates of dengue and chikungunya from previous studies. To ensure adequate statistical power for both dengue and chikungunya, 380 participants were sufficient for dengue (assuming 55% seroprevalence for dengue and 3% for chikungunya). However, this study was a substudy of a larger

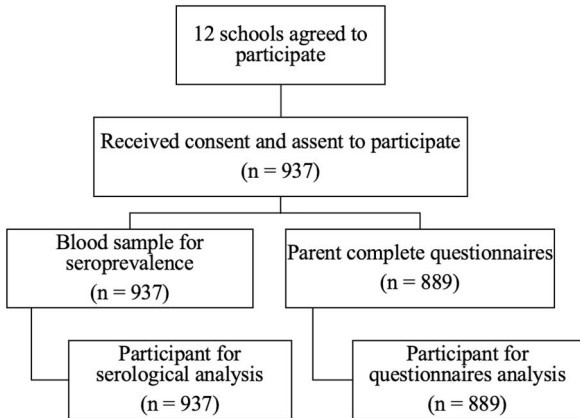

**Fig 1. Study flow chart.**

study, which included at least 840 participants; we enrolled according to the main study. This final sample size provided excellent power for dengue, and still delivered high statistical power for chikungunya.

## Data management and analysis

For each participant, data on age, sex and history of infection were collected. The seroprevalence, defined as a positive IgG result, was calculated overall and stratified by region, with 95% confidence intervals (CI). As the study included only healthy, asymptomatic individuals, isolated IgM-positive results were not included in the analysis due to the potential for cross-reactivity and limited specificity for past infection. The history of dengue vaccination was reported as a percentage. Differences between age groups (10–12 and 13–15 years) and regions within Bangkok (five regions) were analyzed using Chi-square tests and one-way ANOVA. Vaccine knowledge and willingness to vaccinate were presented as percentages. Multivariate regression analysis was conducted using generalized linear models to identify factors associated with vaccine uptake. Data management utilized REDCap version 13.7.25 and was analyzed by SPSS version 29.0.0.0 (241).

## Results

### Demographic data

From June to August 2024, a total of 937 students were enrolled, with a mean age of 11 years (SD: 1.6). The demographic characteristics are summarized in Table 1. Schools were recruited from all regions of Bangkok, including two from the Central, one from the East, one from the North, two from the West, and six from the South.

The questionnaire study included responses from 889 participating parents, representing a 95% response rate. The mean age of respondents was 42 years (SD: 8.8). Most parents (65%) had resided in their current neighborhood for over five years. Regarding housing type, 34% of participants lived in detached houses, 25% in condominiums or apartments, 19% in townhouses, and 17% in other accommodations such as school dormitories or camps. Parental education levels were distributed as follows: 16% had completed primary school, 35% secondary school, and 23% held a bachelor's degree or higher. Additionally, 4% had a vocational certificate (a specific practical training program equivalent to a post-secondary qualification), while 22% did not provide information on educational attainment.

**Table 1. Demographic data and seroprevalence of Dengue and Chikungunya in children aged 10-15 years across Bangkok (N = 937).**

| Demographic data | Number of participants | Dengue | | Chikungunya | |
|---|---|---|---|---|---|
| | n (%) | IgG positive n (%) | p-value | IgG positive n (%) | p-value |
| Age (years) | | | | | |
| 10-12 | 624 (67) | 165 (26) | 0.2 | 38 (6) | 0.73 |
| 13-15 | 313 (33) | 99 (32) | | 21 (7) | |
| Gender | | | | | |
| Male | 496 (53) | 125 (30) | 0.55 | 26 (6) | 0.47 |
| Female | 441 (47) | 117 (31) | | 28 (7) | |
| Bangkok area | | | | | |
| Central | 103 (11) | 34 (33) | 0.10 | 9 (9) | 0.24 |
| North | 94 (10) | 21 (22) | | 2 (2) | |
| South | 520 (55) | 139 (28) | | 36 (7) | |
| East | 64 (7) | 15 (23) | | 2 (3) | |
| West | 156 (17) | 55 (35) | | 10 (6) | |
| Total | 937 (100) | 264 (29) | | 59 (6) | |

## Prevalence of dengue and chikungunya infection

The overall seroprevalence, determined by positive IgG results, was 28.1% (95% CI: 25.2–31.0) for dengue and 6.3% (95% CI: 4.7–7.9) for chikungunya (Table 1 and Fig 2). Dengue IgM positivity was observed in 7% of participants, while chikungunya IgM was positive in 0.2%. None of the IgM-positive participants reported fever at the time of blood collection. Dengue seroprevalence was marginally, but not significantly, higher among older age groups, whereas Chikungunya seroprevalence remained relatively consistent across age groups. Regional variations in seroprevalence were observed, though these differences were not statistically significant. The highest dengue seroprevalence was recorded in the western region of Bangkok (35.3%, 95% CI 32.2-38.4), while the lowest was in the northern region (22.1%, 95% CI 19.4-24.8). Chikungunya seroprevalence was highest in Central Bangkok (8.7%, 95% CI 6.9-10.5) and the lowest in the northern region (2.1%, 95% CI 1.2-3.0).

Thirty-seven participants (3.9%) reported a previous dengue infection confirmed by physicians or blood testing, while four participants (0.4%) reported a prior Chikungunya virus infection. Among those with a documented history of dengue virus infection, 46% tested positive for dengue virus IgG, compared to 27% of those without a reported history of infection. Notably, all four participants with a prior history of Chikungunya virus infection tested positive for Chikungunya virus IgG.

## Knowledge, attitude, and practice regarding dengue and chikungunya

KAP assessments revealed that most parents demonstrated a high level of awareness regarding dengue and chikungunya transmission and symptoms, with 71–94% correctly responding to dengue-related questions and 69–81% correctly answering chikungunya-related items (Table 2). However, only 58% were aware of the availability of dengue vaccines, and just 56 participants (6.6%) reported their child had received it. Of the 56 participants, 32 (57%) received one dose, 19 (34%) received two doses, and 4 (7%) received three doses of the dengue vaccine; one (2%) participant did not report the number of doses received.

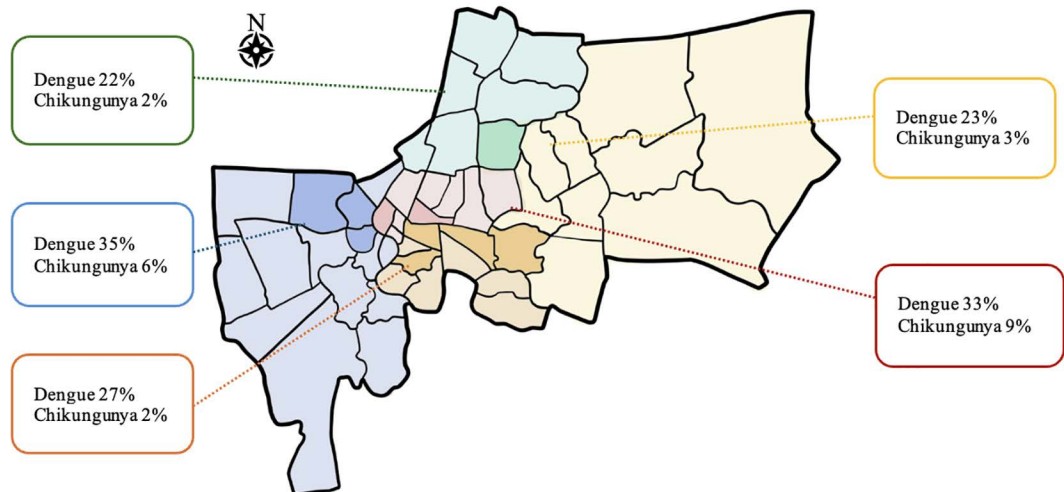

**Fig 2. Seroprevalence of Dengue and Chikungunya by geographic area among Bangkok youth aged 10-15 years (N = 937).** (Each color represents a zone in Bangkok [15]: red for central, green for north, orange for south, yellow for east, and blue for west. Darker shades indicate the study areas where seroprevalence data was collected.).

PLOS Neglected Tropical Diseases

**Table 2. Questionnaires on knowledge of dengue and chikungunya infection (N = 889*).**

| Questionnaires on knowledge of dengue and chikungunya infection | Yes (%) | No (%) |
|---|---|---|
| 1. *Aedes* mosquitoes are carriers of dengue fever | 885 (94) | 4 (1) |
| 2. The rainy season is the period with the highest outbreak of dengue fever | 876 (93) | 13 (2) |
| 3. The mosquitoes that cause dengue fever are active during daytime | 667 (71) | 222 (24) |
| 4. Dengue fever can cause a high fever lasting 2–7 days, along with symptoms like abdominal pain and nausea | 830 (89) | 59 (6) |
| 5. Dengue fever can be transmitted through contact with an infected person | 193 (21) | 692 (74) |
| 6. Currently, there is no vaccine to prevent dengue fever | 344 (37) | 544 (58) |
| 7. Fever, joint pain, and red eyes are symptoms of chikungunya | 762 (81) | 126 (13) |
| 8. In Chikungunya, joint pain can last for months | 651 (69) | 236 (25) |

*The number of parent participants varied depending on the completeness of their questionnaires.

(The correct answers highlighted in green).

Regarding disease perception, 95% of parents recognized the potential severity of dengue, while 91% considered it a significant public health concern. However, 15–20% remained uncertain about the severity and health impact of chikungunya infection (Fig 3A).

With respect to attitudes toward preventive measures, 63% of parents believed mosquito repellent was an effective prevention method, and 91% considered sleeping under mosquito nets as effective. The majority also agreed that community involvement was essential for disease control (Fig 3A). However, actual preventive practices were suboptimal; only one-third regularly checked for standing water, and 10% reported no preventive measures being taken in their residential areas. Additionally, 44% of parents used mosquito repellent before going outside, whereas 34% reported never using it (Fig 3B). Seroprevalence data from children were analysed in relation to preventive measures reported by their parents or guardians. There was no statistically significant difference in seroprevalence between children whose parents reported regularly implementing household preventive measures and those who did not (31%, 95% CI 30.4-30.9 vs 28%, 95% CI 28.0-28.3; p = 0.10).

Most parents reported interest in vaccination, with 96% of parents expressing willingness to receive the dengue vaccine, and 94% were interested in a chikungunya vaccine. All parents of participants with a history of dengue or chikungunya infection were willing to receive the respective vaccines. Among those without a prior history of infection, 96% expressed interest in the dengue vaccination, while 94% indicated willingness to receive a chikungunya vaccination. The primary factors influencing vaccine uptake were safety (65–68%) and efficacy (58–61%). Cost was the least influential consideration, with only 31–34% citing it as a significant concern (Fig 4).

## Discussion

The study provides significant insights into the seroprevalence of dengue and chikungunya among children aged 10–15 years in Bangkok. The observed dengue seroprevalence of 28.1% and chikungunya seroprevalence of 6.3%, as determined by rapid diagnostic tests (RDTs), underscore the continued transmission of these arbovirus infections in Bangkok. Notably, only 4% of participants reported a prior dengue infection, despite the higher measured seroprevalence, suggesting a substantial proportion of asymptomatic or mild cases among children. This finding raises concerns about the risk of severe dengue infections upon secondary infection, emphasizing the need for enhanced surveillance and preventive strategies.

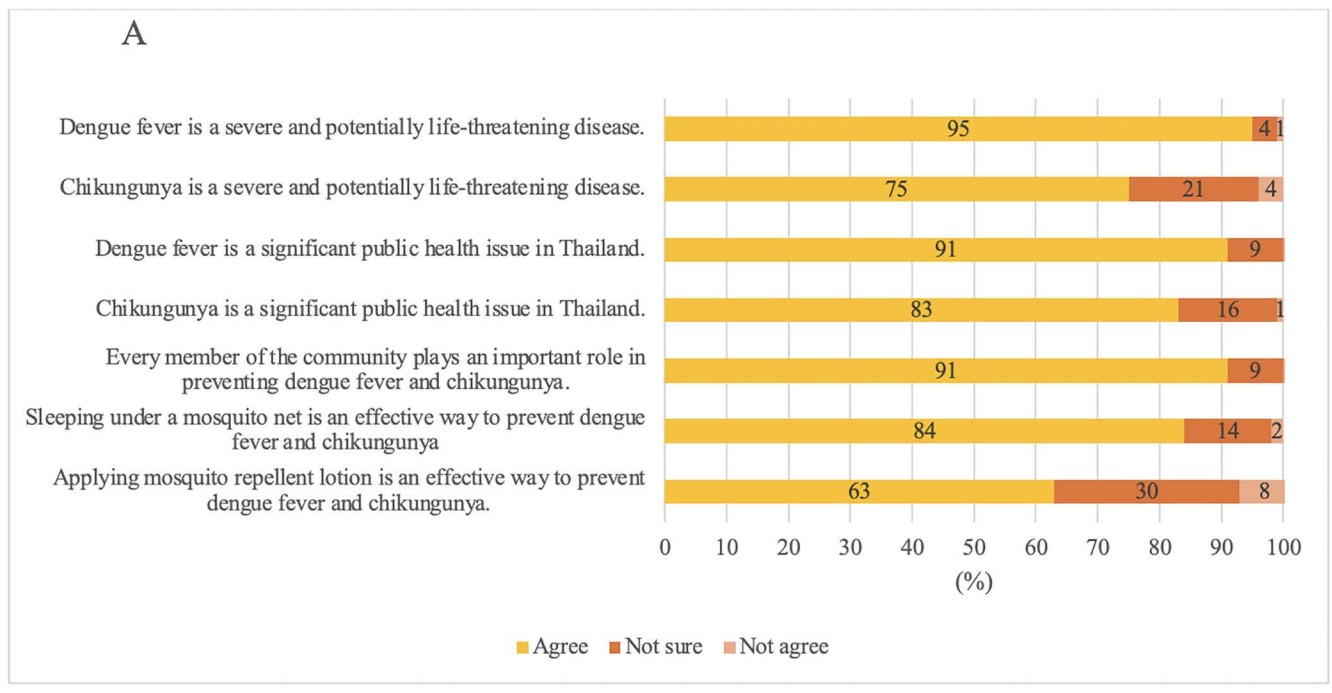

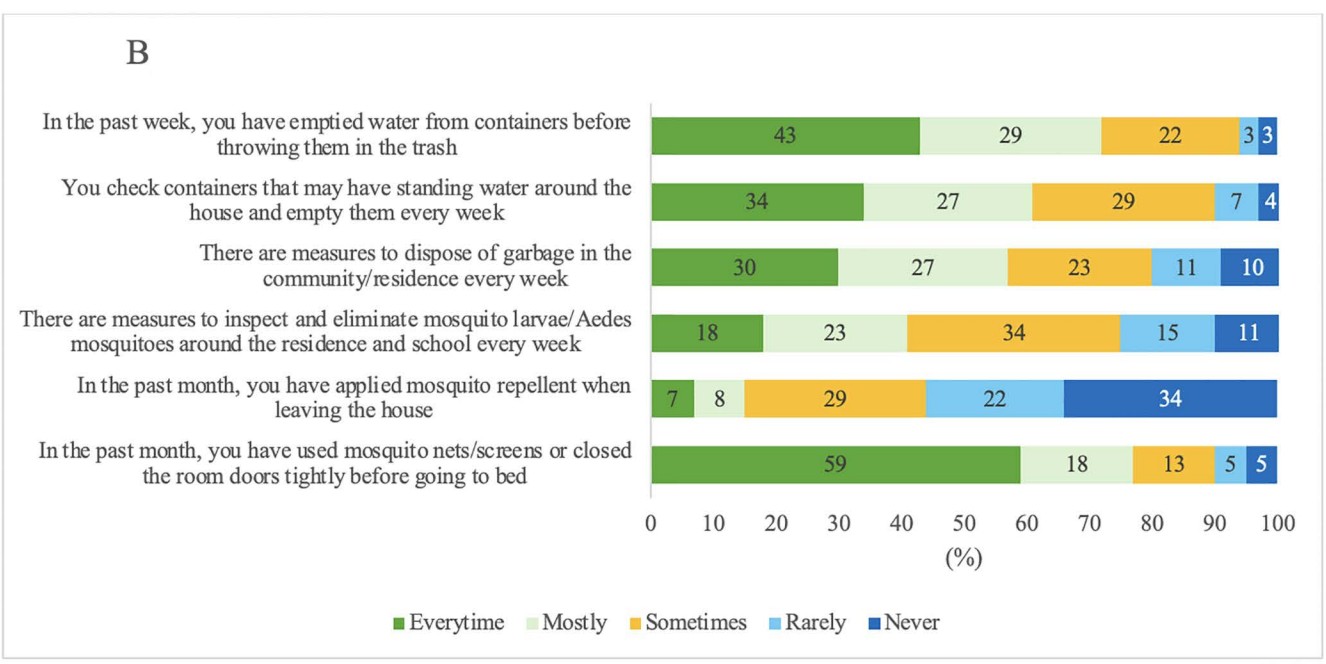

**Fig 3. Questionnaire on Attitudes(Fig 3A) and Practices(Fig 3B) about Dengue and Chikungunya infection and prevention.**

The dengue seroprevalence observed in this study is comparable with seroprevalence by the RDTs but lower than by the ELISA method reported in other parts of Thailand. For example, a 2021 study in Ratchaburi province, an urban center in Western Thailand, found a seroprevalence of 24.3% in the 9–14 year age group using the SD Bioline Dengue Duo

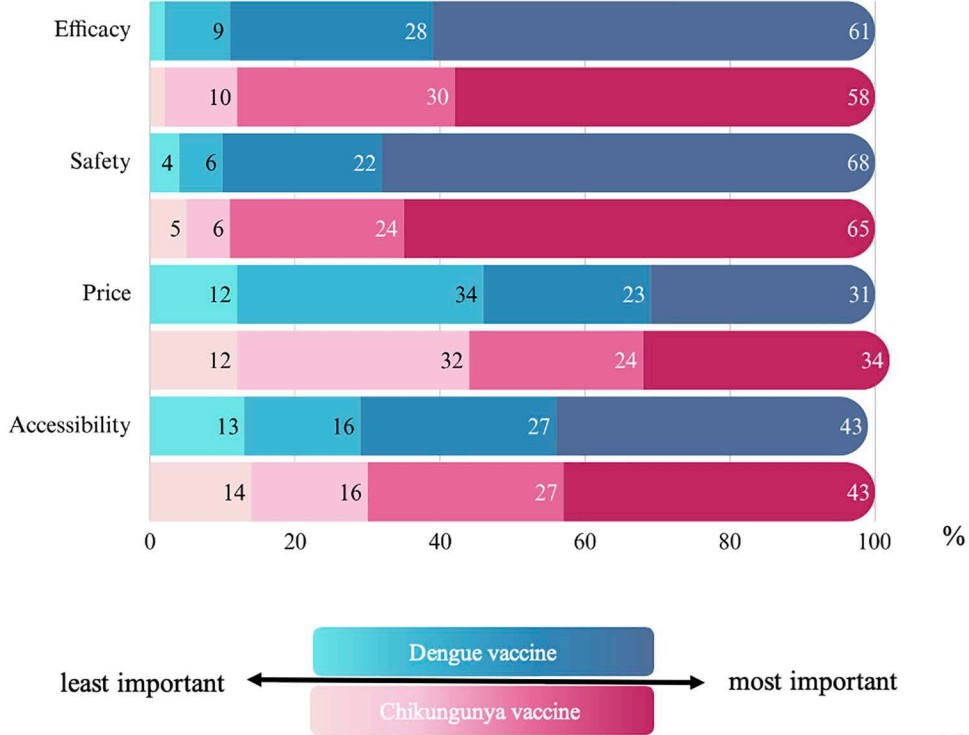

**Fig 4. Factors that influence the reception of dengue and chikungunya vaccines.**

NS1 Ag & IgG/IgM RDT, and 60.5% using ELISA [16]. Similarly, another study in 2023 in 4 urban centers across Thailand (Nonthaburi, Phisanulok, Udon Thani, and Surat Thani Province) estimated a seroprevalence of 55.4% (95%CI 54.3-56.4) in the 10–14 age group using ELISA [17]. Similar seroprevalence rates were observed using ELISA on samples from children (10–15 years of age) in urban and rural sites in North Eastern Thailand [18]. These differences are partly attributable to the use of diagnostic methods with different sensitivities: ELISA is higher sensitivity in detecting past infections compared to the rapid diagnostic test, which primarily identifies acute or recent infections. Such methodological differences, along with variations in local transmission dynamics, may account for the observed discrepancies across studies. Among those who self-reported previous dengue infection, only 46% tested positive for dengue IgG. This discrepancy may reflect waning antibody levels not detectable by the rapid test, recall bias, or a significant proportion of asymptomatic or unrecognized infections in the broader population.

The chikungunya seroprevalence was higher than that reported in a study conducted in 2017, which reported a seropositivity of 3% among individuals aged 10–19 using ELISA [8]. The rising chikungunya seroprevalence in Bangkok may be attributed to the nationwide outbreak in 2019. This outbreak, particularly prominent in Bangkok, was associated with mutations in chikungunya virus strains, such as the E1-A226V mutation, which enhances viral adaptation to *Aedes albopictus* mosquitoes. Previous research suggest that such genetic adaptations contribute to increased transmission and prolonged outbreaks [19,20]. Given these trends, continuous surveillance and research are essential to understanding the epidemiological shifts in chikungunya and informing targeted intervention strategies.

While regional variations in seroprevalence were observed, these differences were not statistically significant, likely due to regional heterogeneity in infection dynamics and sample size limitation. Variability in dengue prevalence across Bangkok aligns with prior findings, where the highest prevalence was reported in the southern region of Bangkok in

2023 [21]. Differences in environmental conditions, population density, and mosquito control measures likely contribute to these disparities, warranting further investigation through spatial epidemiological studies as planned in the larger Geomosquito study.

The KAP assessment demonstrated a high level of awareness regarding dengue and chikungunya among parents, with most recognizing these diseases as significant public health concerns. Knowledge levels were generally consistent across different regions and educational backgrounds, a finding that contrasts with a study in Singapore, where KAP scores varied by education levels [22]. Despite high awareness, gaps in preventive practices were evident. Barriers such as limited use of mosquito repellent, lack of community-wide control initiatives, and socioeconomic constraints may hinder the consistent adoption of preventive behaviors. Additionally, urbanized living environments may make it more difficult for residents to eliminate standing water effectively. Only half of the parents regularly used mosquito repellent, and approximately one-third reported routinely checking for standing water – both crucial practices for mosquito control. The inconsistent use of mosquito repellent may stem from a perceived low risk or forgetfulness.

Despite relatively high awareness of dengue vaccines among the study population, vaccine uptake remained low, with only 7% of children having received at least one dose. Among those vaccinated, most had not completed the full vaccination schedule: 58% received only one dose, 34% received two doses, and 7% received three doses, while one participant did not report the number of doses received. This finding may reflect limited access, vaccine hesitancy, or lack of awareness regarding the recommended three-dose schedule for some dengue vaccines. Although chikungunya awareness was comparatively lower, more than 90% of parents expressed interest in chikungunya vaccination. Current vaccine recommendations primarily target travelers to endemic areas or populations in endemic regions [12]. However, given the rising chikungunya burden in Thailand, post-outbreak seroprevalence studies and consideration of target vaccination strategies are warranted.

Notably, this study found no significant difference in seroprevalence between individuals who adhered to preventive measures and those who did not. This finding underscores the need for additional strategies, such as vaccination, to reduce the disease burden effectively. Although dengue vaccination is effective against severe dengue, the vaccine remains optional in Thailand. Safety and efficacy were the primary concerns among parents, outweighing cost considerations. Given the endemic nature of dengue in Thailand and the exclusion of dengue vaccines from the national immunization program, public health initiatives should focus on expanding vaccine accessibility and targeting high-risk populations to mitigate severe disease outcomes.

The strengths of this study include its large sample size, the specific focus on Bangkok, and an age-specific analysis of children aged 10–15 years, enhancing the reliability and applicability of the findings. In addition to assessing KAP, the study provides insights into vaccination acceptance, which is crucial for informing future vaccination strategies. Furthermore, the study actively promotes vector control awareness by providing education sessions. However, several limitations should be acknowledged. The sampling distribution across Bangkok's geographic zones was not uniform, as school recruitment was based on voluntary participation and operational feasibility. Consequently, certain regions were overrepresented while others contributed fewer samples, which may have introduced bias and contributed to the observed variability in seroprevalence across zones. This uneven representation limits the generalizability of the findings to the broader population of Bangkok or to other regions in Thailand. However, the study does provide preliminary insights into dengue and chikungunya exposure among school-aged children across Bangkok. These findings should be interpreted as exploratory data to inform future, more representative surveillance and intervention planning. Additionally, the use of RDTs for detecting dengue and chikungunya antibodies may have led to an underestimation of true seroprevalence, particularly for past infections where antibody titres decline over time. RDTs typically have lower sensitivity compared to enzyme-linked immunosorbent assays (ELISA) or plaque reduction neutralization tests (PRNT). For example, the Abbott Bioline Dengue IgG/IgM test has been reported to demonstrate a sensitivity of only 21% when compared to ELISA in detecting past dengue infections [23]. Another study in a similar age group in Ratchaburi province compared RDT-based dengue

seroprevalence with ELISA-based data and reported that the ELISA estimate was approximately 2.5 times higher than the RDT estimate, underscoring the substantial difference in detection rates between these methods [15]. ELISA-based studies from 2020 and 2023 reported seroprevalence rates of 59.7%[17] and 55.4%[16], respectively, in similar age groups. Similarly, the Citest Chikungunya IgG/IgM test is designed primarily for use in the acute phase of infection and may exhibit cross-reactivity with antibodies against other flaviviruses. Despite these limitations, the selection of RDTs was based on practical considerations related to field implementation in a school-based setting. RDTs require only a small volume of fingertip blood, are easy to use in non-laboratory environments, and avoid the need for venipuncture, which can be logistically and ethically challenging in school-aged populations. Moreover, in light of the recommendations to use vaccines only in individuals with previous natural exposure to DENV, the use of RDTs will likely be the preferred choice of test as compared to the more laborious use of ELISA. Future studies using more sensitive and specific laboratory assays such as ELISA or PRNT are, however, warranted to obtain more accurate estimates of seroprevalence and confirm these preliminary findings. It is important to note that preventive measures were reported by parents or guardians at the household level and may not fully represent the actual preventive behaviour specifically applied to the children, such as the use of mosquito repellents directly on them. This limitation may affect the accuracy of associations between preventive practices and seroprevalence, and should be considered when interpreting these findings. Future studies could benefit from directly assessing child-specific preventive behaviour.

## Conclusion

Dengue and chikungunya infections remain important public health concerns in Thailand. Our findings suggest that age-related exposure to dengue and early signs of chikungunya transmission are prevalent among school-aged children in Bangkok. While rapid diagnostic tests are limited in sensitivity, particularly for detecting past infections, and may underestimate the true seroprevalence, their ease of use in a school setting provides valuable field-level insights into exposure trends and community preventive practices. These data should be interpreted as exploratory, highlighting the need for more rigorous serological assessments using ELISA or PRNT. The low uptake of dengue vaccination, despite awareness, highlights the need for strengthened public health messaging and improved vaccine accessibility. These findings underscore the importance of integrated community education and vector control strategies to mitigate the burden of mosquito-borne diseases in Thailand.

## Supporting information

**S1 Text. Protocol.**
(PDF)

**S1 STROBE Checklist. von Elm E, Altman DG, Egger M, Pocock SJ, Gøtzsche PC, Vandenbroucke JP; STROBE Initiative.** The Strengthening the Reporting of Observational Studies in Epidemiology (STROBE)statement: guidelines for reporting observational studies. Lancet. 2007 Oct 20;370(9596):1453–7. PMID: 18064739.
(DOCX)

## Acknowledgments

Division of pediatric infectious disease, Department of pediatrics, faculty of medicine, Chulalongkorn university. Center of excellence for pediatric infectious diseases and vaccines, Chulalongkorn university. Affiliated researchers at the faculty of medicine, Chulalongkorn university. Faculty of environmental and resource studies, Mahidol university. Institut de recherche sur l'Asie du Sud-Est contemporaine IRASEC, CNRS, Bangkok. Institut Pasteur, Paris, France. Cassandre von Platen, Institut Pasteur Pole for coordination of clinical research.

## Author contributions

**Conceptualization:** Thitiya Yakasaem, Eric Daudé, Alexandre Cebeillac, Richard Paul, Thanyawee Puthanakit, Watsamon Jantarabenjakul.

**Data curation:** Thitiya Yakasaem, Nattapong Jitrungruengnij, Eric Daudé, Alexandre Cebeillac, Watsamon Jantarabenjakul.

**Formal analysis:** Thitiya Yakasaem, Kanchana Nakhapakorn.

**Funding acquisition:** Richard Paul.

**Investigation:** Thitiya Yakasaem, Padet Siriyasatien, Kanchana Nakhapakorn, Eric Daudé, Alexandre Cebeillac, Richard Paul, Watsamon Jantarabenjakul.

**Methodology:** Thitiya Yakasaem, Eric Daudé, Alexandre Cebeillac, Watsamon Jantarabenjakul.

**Project administration:** Thitiya Yakasaem, Thidarat Jupimai, Napaporn Chantasrisawad, Eric Daudé, Alexandre Cebeillac, Thanyawee Puthanakit, Watsamon Jantarabenjakul.

**Resources:** Thidarat Jupimai, Nattapong Jitrungruengnij, Nattinee Isarankura Na Ayudaya, Paveena Angkhananukit, Pitsamai Ruansil.

**Software:** Kanchana Nakhapakorn.

**Supervision:** Ekasit Kowitdamrong, Padet Siriyasatien, Sunthorn Sunthornchart, Thanyawee Puthanakit, Watsamon Jantarabenjakul.

**Validation:** Watsamon Jantarabenjakul.

**Visualization:** Nattinee Isarankura Na Ayudaya, Pitsamai Ruansil, Thanyawee Puthanakit.

**Writing – original draft:** Thitiya Yakasaem, Watsamon Jantarabenjakul.

**Writing – review & editing:** Richard Paul, Watsamon Jantarabenjakul.

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
