## [Decision Letter · Decision Letter 0]

30 Jun 2025

Seroprevalence and preventive practices of dengue and chikungunya among school children in Bangkok: Gaps in prevention and vaccination strategies

Dear Dr. Jantarabenjakul,

Thank you for submitting your manuscript to PLOS Neglected Tropical Diseases. After careful consideration, we feel that it has merit but does not fully meet PLOS Neglected Tropical Diseases's publication criteria as it currently stands. Therefore, we invite you to submit a revised version of the manuscript that addresses the points raised during the review process.

Please submit your revised manuscript within 60 days Aug 29 2025 11:59PM. If you will need more time than this to complete your revisions, please reply to this message or contact the journal office at plosntds@plos.org. Please include the following items when submitting your revised manuscript:

We look forward to receiving your revised manuscript.

Kind regards,

Adly M.M. Abd-Alla, Prof asso.

Section Editor

Adly Abd-Alla

Section Editor

Shaden Kamhawi

co-Editor-in-Chief

Paul Brindley

co-Editor-in-Chief

**Journal Requirements:**

At this stage, the following Authors/Authors require contributions: Sunthorn Sunthornchart, and Watsamon Jantarabenjakul. Please ensure that the full contributions of each author are acknowledged in the "Add/Edit/Remove Authors" section of our submission form.

- ® on page: 6.

Potential Copyright Issues:

i) Please confirm (a) that you are the photographer of Gravid Ades Trap photo in "2023-136_Geomosquito_protocol_V5-2024-04-04_CL.pdf" file, or (b) provide written permission from the photographer to publish the photo(s) under our CC BY 4.0 license.

ii) Figures 2, and ( figures 1 and 2 in the supplemental file "2023-136_Geomosquito_protocol_V5-2024-04-04_CL.pdf.". Please (a) provide a direct link to the base layer of the map (i.e., the country or region border shape) and ensure this is also included in the figure legend; and (b) provide a link to the terms of use / license information for the base layer image or shapefile. We cannot publish proprietary or copyrighted maps (e.g. Google Maps, Mapquest) and the terms of use for your map base layer must be compatible with our CC BY 4.0 license.

6) In the online submission form, you indicated that "The data that support the findings of this study are available from the corresponding author upon reasonable request." All PLOS journals now require all data underlying the findings described in their manuscript to be freely available to other researchers, either

1. In a public repository

2. Within the manuscript itself

3. Uploaded as supplementary information.

7) Please amend your detailed Financial Disclosure statement. This is published with the article. It must therefore be completed in full sentences and contain the exact wording you wish to be published.

8) Thank you for indicating that "The authors declare that they have no known competing financial interests or personal relationships that could have appeared to influence the work reported in this paper." Please state "The authors have declared that no competing interests exist".

**Reviewers' Comments:**

Reviewer's Responses to Questions

**Key Review Criteria Required for Acceptance?**

**Methods**

-Are the objectives of the study clearly articulated with a clear testable hypothesis stated?

-Is the study design appropriate to address the stated objectives?

-Is the population clearly described and appropriate for the hypothesis being tested?

-Is the sample size sufficient to ensure adequate power to address the hypothesis being tested?

-Were correct statistical analysis used to support conclusions?

-Are there concerns about ethical or regulatory requirements being met?

Reviewer #1: The objectives of the study are clearly articulated, aiming to assess the seroprevalence of dengue and chikungunya infections among children aged 10-15 years in Bangkok and evaluate the knowledge, attitudes, and practices (KAP) related to mosquito-borne disease prevention. However, the hypothesis could be more explicitly stated.

The cross-sectional descriptive study design is appropriate for addressing the stated objectives. However, the reliance on rapid diagnostic tests (Abbott DENGUE IgG/IgM and Citest Chikungunya IgG/IgM) raises concerns about the accuracy of the seroprevalence data.

The population is clearly described, including children aged 10-15 years from 12 schools in Bangkok. The sample size of 937 participants appears sufficient to ensure adequate power for dengue but is close to the required size for chikungunya.

The statistical analyses used to support the conclusions are appropriate. However, the limitations of the diagnostic tests used should be considered in the interpretation of the results.

Ethical and regulatory requirements appear to be met, with approvals from relevant institutional review boards and informed consent obtained from participants and their parents.

Reviewer #2: • Sample size calculation: Details provided here are good for a proposal but not necessarily so for a scientific manuscript. You may need to summarize it.

Reviewer #3: The methodology used for sampling have a serious error on the distribution pattern of the samples selected. The study was carried out in 5 zones of Thailand viz., Northern, Southern, Eastern, Western, and Central. However, authors chose 12 schools agreed to participate from these sites one, six, one, two, six and two respectively. As a result, the samples collected from these regions were also show wide variations in seroprevalence data for Dengue. Hence, the results obtained in the study may have some range of ambiguity, when generalized for the Country. (Authors state these as limitations in the study).

**Results**

-Does the analysis presented match the analysis plan?

-Are the results clearly and completely presented?

-Are the figures (Tables, Images) of sufficient quality for clarity?

Reviewer #1: The analysis presented matches the analysis plan. However, the results may be compromised by the limitations of the rapid diagnostic tests used.

The results are clearly and completely presented, with demographic data, seroprevalence rates, and KAP assessments detailed. Figures and tables are of sufficient quality for clarity.

The figures and tables are of sufficient quality and clarity to illustrate the study's findings.

Reviewer #2: • Table 1: The kit picks IgG/IgM, so why is only IgG being reported here?

• Prevalence of dengue and chikungunya infections (line 218-219): As this was a cross-sectional study, trends over time cannot be assessed. The statement should be revised to reflect a single point-in-time observation rather than implying a trend.

• Knowledge, attitude and practice regarding dengue and chikungunya (line 245): How many doses of the vaccine did they receive?

• Line 272-274: Seroprevalence was measured in school children, while information on preventive measures was collected from parents. How were the two datasets linked to support the inferences made? For example, you mentioned that some parents use mosquito repellents before going out, but it was not specified whether these repellents were also used on their children. These assumptions need to be clarified to strengthen the validity of this statement.

Reviewer #3: Results are presented well

**Conclusions**

-Are the conclusions supported by the data presented?

-Are the limitations of analysis clearly described?

-Do the authors discuss how these data can be helpful to advance our understanding of the topic under study?

-Is public health relevance addressed?

Reviewer #1: The conclusions are not supported by the data presented. There are no limitations of the diagnostic tests described and the significant concerns regarding their accuracy for seroprevalence studies should be more thoroughly addressed.

The authors discuss how the data can advance understanding of dengue and chikungunya seroprevalence and preventive practices. However, the potential inaccuracies in the seroprevalence data may limit the study's contributions.

The public health relevance is addressed, emphasizing the need for enhanced community education and vector control interventions.

Reviewer #2: Discussion

• Line 302: ‘arbovirus’, correct the spelling

• Line 312-315: Misleading interpretation:

• If only 46% of those self-reporting past dengue tested positive, it doesn't mean seroprevalence is higher—it may actually suggest asymptomatic infections or recall bias.

• Conversely, if you're suggesting that many who didn’t report previous dengue still had IgG antibodies, that could suggest underreporting and higher seroprevalence—but this isn't what the sentence says. Rephrase for clarity and correctness

• Line 322: ‘research’, it has no plural form

• Line 338: The questionnaire assessed the ‘use’ of repellents, not ‘access’ to them. Please revise the wording accordingly to accurately reflect what was measured.

• Line 347: Clarify if the children took one or two doses of the vaccine.

Conclusion

• Line 381-382: Data provided does not support this assumption; hence, it is advisable to leave it out of your conclusion.

Reviewer #3: As mentioned above as the study does not follow a statistically sound procedure on the distribution of samples, the results obtained may not be generalized for the Country. Besides, the specificity and sensitivity of the IgG kits used to estimate the seroprevalence of Dengue and Chikungunya may have to be justified by previous studies carried out using these kits if any. Authors have provided only references of the firms that manufactured the kits.

**Editorial and Data Presentation Modifications?**

Reviewer #1: (No Response)

Reviewer #2: None

Reviewer #3: Discussion: Line 321 Aedes aegypti may be corrected as Aedes albopictus, as E1 A226V mutation provides transmission advantage to the latter species

**Summary and General Comments**

Reviewer #1: I have carefully reviewed the article titled "Seroprevalence and preventive practices of dengue and chikungunya among school children in Bangkok: Gaps in prevention and vaccination strategies" submitted for publication in PLOS Neglected Tropical Diseases.

The study aims to assess the seroprevalence of dengue and chikungunya infections among children aged 10-15 years in Bangkok and evaluate the knowledge, attitudes, and practices (KAP) related to mosquito-borne disease prevention. The methodology involves a cross-sectional descriptive study conducted across 12 schools in Bangkok, using rapid diagnostic tests (Abbott DENGUE IgG/IgM and Citest Chikungunya IgG/IgM) based on the immunochromatography technique.

This article presents potentially valuable information because it aims to address a significant public health concern in Thailand, focusing on the seroprevalence of Chikungunya and dengue in vaccine eligible school children. The inclusion of KAP assessments provides valuable insights into community awareness and preventive practices. In addition, the study's sample size (937 participants) appears to be adequate to support the findings for Dengue and very close to those required for Chikungunya.

However, I have the following major concerns with this manuscript:

The primary concern with this study is the reliance on rapid diagnostic tests (Abbott DENGUE IgG/IgM and Citest Chikungunya IgG/IgM) to determine seroprevalence. These tests are designed for the qualitative detection of anti-DENV IgM and IgG antibodies to aid in distinguishing acute primary versus secondary infections. However, seroprevalence studies require accurate, quantitative (or at least highly sensitive) measurement of past exposure across all age groups. Several key limitations prevent the Bioline RDT from reliably estimating true IgG seroprevalence:

• The Abbott Bioline Dengue IgG/IgM has low field sensitivity for detecting past (IgG) infections: Studies such as Arkell P, et. al. (2022). Field evaluation of rapid diagnostic tests to determine dengue serostatus in Timor-Leste. PLoS Negl Trop Dis 16(11): e0010877, have shown that the Bioline IgG/IgM test has low sensitivity (21.1%) for detecting past IgG infections compared to reference IgG ELISA tests.

• The Abbott Bioline Dengue IgG/IgM and the Citest Chikungunya IgG/IgM cassette were optimized for the detection of acute-phase antibodies: RDTs are optimized to detect high-titer IgM and acute/secondary IgG boosts, rather than low, stable IgG levels months to years after infection. This can result in false-negative results for individuals with remote infections. The use of rapid diagnostic tests, which have lower sensitivity compared to ELISA or PRNT, may have underestimated seroprevalence, particularly for past infections where antibody levels decline over time. No data was presented in this manuscript demonstrating that the test was validated for determining seroprevalence of past infections with more sensitive methods.

• Potential Cross-Reactivity: Cross-reactivity with other flaviviruses can lead to misclassification and should be addressed.

While the study provides valuable insights into the seroprevalence of dengue and Chikungunya among school children in Bangkok, the use of rapid diagnostic tests with known limitations for seroprevalence studies raises concerns about the accuracy of the findings. For reliable estimation of age-specific or population-level dengue seroprevalence, laboratory-based IgG ELISAs or Plaque Reduction Neutralization Tests (PRNT) remain the gold standards. Therefore, I recommend reconsidering the publication of this study until more accurate diagnostic methods are employed to validate the seroprevalence data.

Reviewer #2: This manuscript offers valuable and novel insights into the topical issue of dengue and chikungunya infections among school children in Bangkok. It also explores parental knowledge, attitudes, and practices, uncovering findings with potential implications for policy and improved disease control strategies in the region. However, the manuscript would benefit from thorough English language editing to address grammatical and stylistic issues. Overall, this work makes a meaningful contribution to the body of knowledge in this field.

Reviewer #3: Sampling errors and methodological ambiguity exist in the study. Hence the results may not represent the true picture of seroprevalence of Dengue and Chikungunya in Thailand.

PLOS authors have the option to publish the peer review history of their article (what does this mean? ). If published, this will include your full peer review and any attached files.

**Do you want your identity to be public for this peer review?** For information about this choice, including consent withdrawal, please see our Privacy Policy .

Reviewer #1: No

Reviewer #2: No

Reviewer #3: **Yes:** N. Pradeep Kumar

**Figure resubmission:**

**Reproducibility:**



---

## [Decision Letter · Decision Letter 1]

4 Mar 2026

Dear Assist. Prof. Dr. Jantarabenjakul,

We are pleased to inform you that your manuscript 'Seroprevalence and preventive practices of dengue and chikungunya among school children in Bangkok: Gaps in prevention and vaccination strategies' has been provisionally accepted for publication in PLOS Neglected Tropical Diseases.

Best regards,

Amy C. Morrison, PhD

Section Editor

Amy Morrison

Section Editor

Shaden Kamhawi

co-Editor-in-Chief

Paul Brindley

co-Editor-in-Chief

Reviewer's Responses to Questions

**Key Review Criteria Required for Acceptance?**

**Methods**

-Are the objectives of the study clearly articulated with a clear testable hypothesis stated?

-Is the study design appropriate to address the stated objectives?

-Is the population clearly described and appropriate for the hypothesis being tested?

-Is the sample size sufficient to ensure adequate power to address the hypothesis being tested?

-Were correct statistical analysis used to support conclusions?

-Are there concerns about ethical or regulatory requirements being met?

Reviewer #1: (No Response)

**Results**

-Does the analysis presented match the analysis plan?

-Are the results clearly and completely presented?

-Are the figures (Tables, Images) of sufficient quality for clarity?

Reviewer #1: (No Response)

**Conclusions**

-Are the conclusions supported by the data presented?

-Are the limitations of analysis clearly described?

-Do the authors discuss how these data can be helpful to advance our understanding of the topic under study?

-Is public health relevance addressed?

Reviewer #1: (No Response)

**Editorial and Data Presentation Modifications?**

Reviewer #1: (No Response)

**Summary and General Comments**

Reviewer #1: The authors have successfully addressed my comments.

PLOS authors have the option to publish the peer review history of their article (what does this mean? ). If published, this will include your full peer review and any attached files.

**Do you want your identity to be public for this peer review?** For information about this choice, including consent withdrawal, please see our Privacy Policy .

Reviewer #1: No

---

## [Editor Report · Acceptance letter]

Dear Assist. Prof. Dr. Jantarabenjakul,

We are delighted to inform you that your manuscript, "Seroprevalence and preventive practices of dengue and chikungunya among school children in Bangkok: Gaps in prevention and vaccination strategies," has been formally accepted for publication in PLOS Neglected Tropical Diseases.

Best regards,

Shaden Kamhawi

co-Editor-in-Chief

Paul Brindley

co-Editor-in-Chief
